# Contest-Based and Norm-Based Interventions: (How) Do They Differ in Attitudes, Norms, and Behaviors?

**Magnus Bergquist \*, Andreas Nilsson and Emma Ejelöv**

Department of Psychology, University of Gothenburg, SE-412 96 Gothenburg, Sweden;
andreas.nilsson@psy.gu.se (A.N.); emma.ejelov@psy.gu.se (E.E.)

\* Correspondence: magnus.bergquist@psy.gu.se

**Abstract:** Setting up a contest is a popular means to promote pro-environmental behaviors. Yet, research on contest-based interventions is scarce while norm-based interventions have gained much attention. In two field experiments, we randomly assigned 79 apartments to either a contest-based or a norm-based electricity conservation intervention and measured kWh usage for 2 and 4 weeks, respectively. Results from both studies showed that contest-based interventions promote intensive but short-lived electricity saving. In Study 1 apartments assigned to a norm-based intervention showed more stable electricity saving (low intensity and long-lasting). Study 2 did not replicate this finding, but supported that participants in the norm-based intervention also engaged in non-targeted behaviors. These results emphasize the importance of identifying how different intervention techniques may activate different goals, framing both how people think about and act upon targeted pro-environmental behaviors.

**Keywords:** behavioral intervention; household energy demand; norms; contest; goals; attitudes

---

## 1. Introduction

Household energy use can be decreased by 20% through simple changes in everyday behaviors [1]. How should energy conservation interventions be designed to motivate such behavioral change? In general, interventions promoting behaviors that benefits the environment or harms the environment as little as possible [2] (i.e. pro-environmental interventions), seek to promote behavior change by appealing to various motives [3], such as improving one's economy [4,5], avoiding health-related risks [6], or conforming to social norms [7]. These motives all have the capacity to change behaviors, but they may also frame different goals, affecting *how* people think and act pro-environmentally [3]. For example, interventions motivating behavioral change through monetary incentives have shown to both decrease peoples motivation [8] and perceived responsibility [9,10]. Furthermore, monetary incentives have also shown to prevent future engagement in the targeted behavior [11]. In contrast, interventions based on social feedback have shown to foster pro-social behaviors [12] which may spread to additional pro-social behaviors [13,14]. The present study examines both behavioral and psychological effects of contest-based and norm-based intervention techniques targeting electricity conservation.

Contest-based interventions involve incentivizing pro-environmental behavior change by offering a reward for the person or group most successful in their behavior change. The contest is thus a form of incentive. Contest-based interventions aiming to change behaviors are popular among practitioners. For example, large-scale intervention campaigns such as "Student Switch Off" [15] assume that energy conservation behaviors can be fostered by contests. Yet, the contest-based intervention is underexplored by researchers [16,17]. Although past studies on contest-based interventions have shown that contests

can promote energy conservation [18–21], the resulting behavioral change seems to be less long-lasting than in norm-based interventions [18,21–25].

The norm-based interventions make use of and communicate via social feedback about other peoples´ behaviors and (dis)approvals to reduce, for example, water and energy consumption [7,26–29]. Research shows that the norm-based intervention is most effective when it includes two types of norms: descriptive norms and injunctive norms [7,30–32]. These two norms complement each other as descriptive norms simply describe others' behaviors (e.g., "Over XX% of households in your community use effective landscape irrigation techniques" [26]), while the injunctive norm are informing about others´ approval or disapproval of that specific behavior (e.g., adding a happy or sad emoticon to signal others (dis)approval of a specific behavior [7]). These two norm concepts are not always aligned, for example, people may believe that conserving energy is an approved behavior but still not performed by others. Messages that convey these norms are stronger when the two norm concepts are aligned and thus support one another [7]. Two explanations for why people follow social norms are that people want to act correctly or effectively, and that people want to gain others' approval [32–34]. These motives differ quite substantially to the financially based motives in contest-based interventions. Thus, acting pro-environmentally to win a contest or to follow social norms may affect both how people think and feel about a behavior, and also how people go about in conducting a specific behavior.

In comparing the contest-based to the norm-based intervention, we adopted a goal-based perspective, examining how these two interventions differ in terms of the psychological goals. A distinction between psychological goals is important as goals have different implications for peoples´ motivation and behavioral [35–37]. For example, if behavioral change are long-termed and if the interventions also affects non-targeted pro-environmental behaviors. In brief, we suggest that a norm-based intervention is a form of normative goal frame, making people perceive and act upon the targeted behavior in term of perceived obligations, while the contest-based intervention is a form of gain goal frame, promoting instrumental behavioral change to improve financial resources.

The article is outlined in the following way, first we provide a theoretical rational for the different effects of a norm-based and contest-based intervention, we then provide a description of the method and a combined results and discussion section for Study 1. We then move on to describe the method and results for Study 2 (see Appendix A for summary of the experimental designs). We conclude with a general discussion of the findings.

**Theoretical Framework.** Human motivation and behavior are not stable across situations (e.g., [38]). An individual may, in some situations, seek to profit by rationally weighing benefits against losses, while in other situations, act on collective norms by conforming to other people's behavior. How can such diversity in motivation and decision-making styles be explained? Goal Framing Theory (GFT) [35–37] builds on findings suggesting that humans cognitive capacity is limited: people simply cannot attend to all aspects or information in a given situation. Rather, situational goals will influence how individuals think (more specifically, how people detect information, construct preferences, and perceive behavioral alternatives) [39,40]. GFT proposes three such overarching goals: the gain goal, the normative goal, and the hedonic goal. When framing a *gain goal*, people will be motivated by, and think in terms of improving or guarding their resources. In contrast, the *normative goal* sensitizes people to appropriate behavior, for example what one "ought to" do. Finally, the *hedonic goal* makes people think and act in terms of seeking pleasure and avoiding pain. All goals have the capacity to influence behavior, but situational cues will bring a particular goal to the foreground of attention while pushing other goals to the background [36]. As a concrete example of such a goal-framing process, GFT has been applied to understand how socially disapproved behaviors spreads. Keizer and colleagues [41] found that norm-violations such as painting graffiti and setting off fireworks led to additional norm-violations such as littering and trespassing, as cues of norm-violations are predicted to weaken the relative strength of the normative goal and strengthen the impact of gain and/or hedonic goals. In this paper, we propose that the contest-based intervention is a form of gain goal frame, while the norm-based intervention is a form of normative goal frame. Consequently, we predict that

contest- and norm-based interventions differ both in psychological and behavioral implications. These differences will be further elaborated on below.

The gain goal can be framed by money, for example in a poker game or by sales in a boutique [42]. Such situational gain cues have shown to increase behavioral engagement while decreasing normative considerations [38,41–43]. Applied to pro-environmental interventions, thinking about the economic aspects of energy conservation may lead people to conserve energy as a means to an end (i.e., saving energy to save money). Therefore, we expect participants in the contest-based intervention to perceive electricity conservation in terms of a gain-goal, and thus engage in electricity conservation as a way to increase their own resources. This will promote intensive behavioral engagement, but only for as long as the engagement is associated with the possibility of winning a prize. In testing the psychological effects of contest, past research has found that contests increase unethical behavior [44], plausibly because the contest pushes away normative considerations. We therefore expect that contest-based interventions will promote electricity conservation without affecting attitudes (i.e., general positive or negative evaluations towards an attitude object, for example, if participants view electricity conservation as meaningless or meaningful, unimportant or important, and bad or good) and normative considerations (i.e., internalized feelings of obligation, for example, if participants' feel that save electricity is "the right thing to do"). Although these concepts overlap, the key difference is that while attitudes measure people's evaluations, normative considerations is intended to capture people's moral beliefs.

The normative goal can be framed by seeing other people picking up litter or sweeping the street [13,40,45] (i.e., by perceiving or being informed about other peoples' moral behavior). We propose that normative feedback about other peoples' approval of and engagement in electricity conservation is a form of normative goal frame [37]. Drawing on GFT, we therefore expect that a normative goal will positively affect peoples' attitudes and normative considerations [40]. That is, we seek to test both peoples own opinions about conserving electricity (e.g., if they view a specific conservation behavior in a positive or negative manner) and normative considerations (e.g., whether people perceive that it is right to conserve electricity). We further expect that the psychological effects of the norm-based intervention will promote more long-lasting electricity conservation (see also [36,40] for discussion on the normative goal and intrinsic motivation).

Finally, signs that strengthen the normative goal have been shown to increase subsequent behaviors [13,41] The potential for one behavior to increase the likelihood of performing a subsequent behavior are examined within the spillover literature (see [46,47], for review). As spillovers can be either positive (i.e., where a first pro-environmental actions leads to subsequent pro-environmental actions) or negative (i.e., where a first pro-environmental actions hinders subsequent pro-environmental actions), it is important to assess spillovers within interventions, as potential spillover effects provides a more comprehensive picture of the intervention [48,49]. We will examine spillover effects between electricity conservation and other types of conservation behaviors in Study 2.

## 2. Study 1

### 2.1. Hypotheses

The aim of Study 1 was to compare a contest-based intervention to a norm-based intervention. In line with GFT [3,40] and research on pro-environmental interventions [19,50], we expected that a contest-based intervention would promote behavioral change without increasing attitudes or normative considerations about electricity conservation and that electricity conservation behaviors would last only for as long as the goal of winning a prize remained salient (i.e., only during the intervention period). For the norm-based intervention we expected that the normative goal frame would foster both positive attitudes and personal norms, leading participants to perceive electricity conservation as "the right thing to do." We further predicted that activating these attitudes and normative considerations would lead to more long-lasting electricity conservation in the norm-based intervention than in the contest-based intervention. We therefore set out to investigate the following hypotheses:

**H1.** *Participants in the contest-based intervention condition will show more intensive but shorter-lived electricity saving than participants in the norm-based intervention condition.*

**H2.** *Participants in the norm-based intervention condition will show less intensive but more long-lasting electricity saving than participants in the contest-based intervention condition.*

**H3.** *Participants in the norm-based intervention condition will show more increase in personal electricity conservation norms than participants in the contest-based intervention condition.*

**H4.** *Participants in the norm-based intervention condition will show more increase in positive electricity conservation attitudes than participants in the contest-based intervention condition.*

*2.2. Method*

**Procedure.** Residents of 150 one-room student apartments in Gothenburg, Sweden, in two separate, identical apartment blocks, were offered the opportunity to participate in an electricity conservation study. Participants were recruited via door hangers stating that the study would run for two weeks and that all participants would receive a lottery ticket (as an incentive to participate) and a refrigerator magnet with electricity saving tips. The prize for participants in the contest-based intervention was thus not disclosed at the recruiting stage. Participants were informed that their participation constituted signed consent for access to their electricity data. Furthermore, all subjects were informed that participation was anonymous, that they had the rights to end their participation, and that the data would be used for research purposes only. Within a week of distribution, the door hangers were collected and the two apartment blocks were randomly assigned to either a contest-based or a norm-based intervention. All participants received two surveys: a pre-intervention survey e-mailed before the intervention started and a post-intervention survey e-mailed the day the intervention ended (see Table A1 for overview of the experimental design).

**Participants.** A final sample of 19 participants (53% females, $M_{age}$ = 23.63, *SD* = 2.41, recruitment rate = 12.67%) were randomly assigned to either a contest-based or a norm-based intervention targeting electricity conservation. We measured participants' electricity conservation attitudes, electricity conservation norms, and electricity usage. Apartment electricity data was obtained two weeks prior to, during, and two weeks after the intervention (1st September to 15th October 2016). The study followed ethical guidelines in Sweden for survey data and were thus conducted in line with the declaration of Helsinki

**The Two Interventions**. During the intervention, contest-based or norm-based information was provided to participants via posters, a refrigerator magnet, and in one e-mail. In each apartment block, posters (140 × 100 cm) informing about either a social norm or a contest were put up in a visible space near the residents' post boxes. In the norm intervention, posters communicated a descriptive norm by stating that other students on their street engaged in electricity conservation and an injunctive norm presenting electricity conservation as an appropriate collective behavior. The norm poster included human figures cooperating to save electricity and the text "Do like the rest of us! Together we'll save electricity!". In the contest intervention, the posters presented electricity conservation as a means to save and make money, and the contest poster included human figures in competition, signs of money, and a text reading "Be the winner! Save the most electricity of all!". These manipulations were also printed on refrigerator magnets given to all apartments and attached with the e-mailed the pre-intervention survey. In addition, the contest group was told that the student that had saved the most electricity by the end of the study would win a prize (200 SEK). The prize was later rewarded to the winner.

**Measures.** Both the pre- and post-intervention surveys contained questions assessing attitudes toward electricity-conserving behavior and personal norms. Attitudes were assessed with 8 items (e.g., "What is your opinion on turning off the light when you leave the room?"), all measured on a 7-point Likert scale (1 = "extremely negative" to 7 = "extremely positive"; *M* = 5.71, *SD* = 0.73).

Personal norms for conserving electricity were assessed with one item, "Do you feel a strong personal obligation to save electricity?" measured on a 7-point Likert scale (1 = "very little" to 7 = "very much"; $M = 5.47$, $SD = 1.17$). These items were included in both the pre- and post-intervention surveys. The post-intervention survey also contained control questions (e.g., whether participants had put up the refrigerator magnet during the intervention and had seen the norm and/or contest posters).

## 2.3. Results and Discussion

**Control questions and reliability analysis.** Of the 22 recruited participants, 21 answered both surveys and 2 were excluded, as one participant was an outlier on several variables and the other lived in a three-room apartment. The final sample therefore included 19 participants ($n_{contest} = 11$; $n_{norm} = 8$). Of the 19 participants, 17 reported putting up the refrigerator magnet ($n_{contest} = 11$, $n_{norm} = 6$), and 18 had seen the posters during the intervention ($n_{contest} = 11$, $n_{norm} = 7$), and in the contest intervention 5/11 reported the correct prize in the contest. During the intervention, 10/19 did not have visitors ($n_{contest} = 6$) and 10/19 had not been out of town ($n_{contest} = 4$). Analysis found no significant correlations between electricity usage and number of visitors or electricity usage and number of days out of town (all $p > 0.05$), suggesting that these variables did not affect the kWh data. As a manipulation check, participants were asked how cooperative or competitive they felt when seeing the contest/norm manipulation. Only nine participants answered this question ($n_{contest} = 3$, $n_{norm} = 7$). The data showed a medium effect size, confirming that participants in the norm intervention felt more cooperative ($d = 0.29$) and participants in the contest intervention felt more competitive ($d = 0.29$). These differences were not statistically significant. A Cronbach's alpha analysis showed acceptable reliability values for attitude toward electricity conservation in the pre-measure ($\alpha = 0.80$) but weak reliability in the post-measure ($\alpha = 0.64$).

**Main Analysis.** We first conducted a 2 (intervention: contest versus norm) × 4 (kWh usage: pre-intervention versus first week of the intervention versus second week of the intervention versus post-intervention) mixed analysis of variance (ANOVA) with repeated measures on the last condition testing for kWh usage. A sensitivity power analysis, computed at $a = 0.05$, $1 - b = 0.80$, $r_{repeated\ measures} = 0.83$, $e = 1$, showed that the analysis was adequately powered to detect an effect size of $f = 0.16$. It revealed a significant main effect for kWh usage, ($F(3, 17) = 7.64$, $p < 0.001$, $\eta_p^2 = 0.31$, see Table 1 and Figure 1) showing that kWh usage decreased in both interventions. To examine the interaction between electricity usage and intervention technique, that is if electricity usage differed between the two interventions, we performed a simple slopes analysis. For the contest intervention, kWh usage was significantly lower than baseline for the first week of the intervention ($p < 0.001$), but only marginally lower for the second week of the intervention ($p = 0.09$). In the post-intervention period, kWh was significantly lower than baseline ($p = 0.04$). This implies that the intensive behavior change predicted by the contest incentive may be even more short-lived than previously assumed. In the norm intervention, kWh usage was significantly lower than baseline for both weeks of the intervention and during the post-intervention period ($p < 0.05$). Moreover, we conducted within-subject contrasts to examine if the trend in electricity usage for each intervention was best explained by a linear trend or by a curve-linear trend. For the contest intervention, we found a significant quadratic trend ($F(1, 10) = 9.69$, $p = 0.01$) (Friedman non-parametric test found a significant main effect in the contest-based intervention ($p = 0.003$), and a marginally significant main effect in the norm-based intervention ($p = 0.070$)). As can be seen in Figure 1, these results support H1, showing that electricity conservation was intensive but short-lived in the contest-based intervention. For the norm intervention, we found a significant linear trend ($F(1, 7) = 5.7$, $p = 0.048$) supporting our prediction that participants in the norm-based intervention would show less intensive and longer-lasting electricity saving. Taken together, although the data suffered from fairly low power due to small sample size, the effect size and the trend analysis are in line with our hypotheses, suggesting differences in behavioral implications between the contest- and norm-based interventions.

**Table 1.** Means and standard deviations for kWh usage in Study 1.

|  | **Pre-Intervention** | **Week 1** | **Week 2** | **Post-Intervention** |
|---|---|---|---|---|
| **Contest-based intervention** | 4.00 (0.86) | 3.19 (1.10) | 3.6 (1.12) | 3.61 (1.04) |
| **Norm-based intervention** | 4.15 (1.24) | 3.63 (0.98) | 3.59 (1.06) | 3.55 (1.35) |

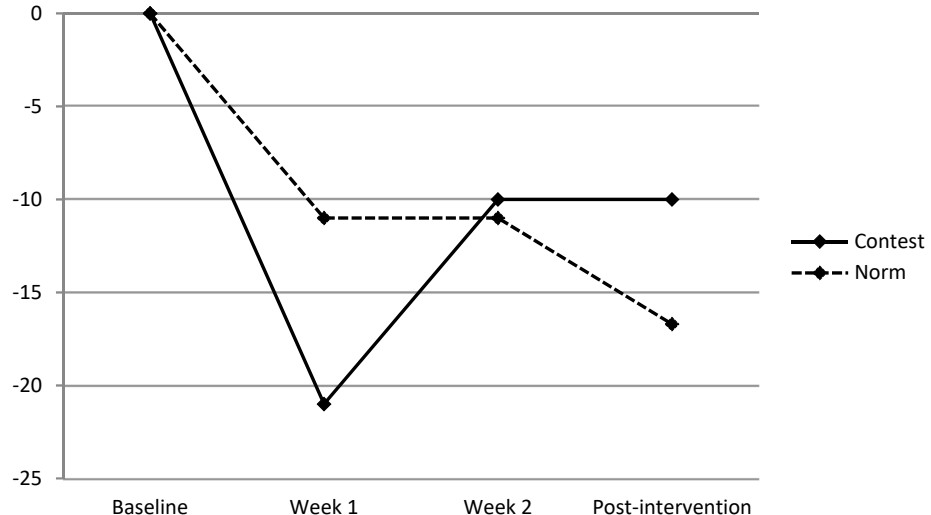

**Figure 1.** Percentage changes in kWh use from baseline to the first and second week of the intervention and during the post-intervention period for the contest-based and norm-based interventions.

To test H3, that personal electricity conservation norms would increase more in the norm-based intervention than in the contest-based intervention, we conducted a 2 (intervention: contest versus norm) × 2 (personal norms: pre- versus post-measure) mixed ANOVA with repeated measures on the last condition for ratings of personal electricity conservation norms. A sensitivity power analysis, computed at $a = 0.05$, $1 - b = 0.80$, $r_{repeated\ measures} = 0.69$, $e = 1$, showed that the analysis was adequately powered to detect an effect size of $f = 0.27$. Results showed no significant main or interaction effects (all $F's < 1$). Hypothesis 3 was thus not supported.

Finally, to test H4, that electricity conservation attitudes would show a stronger increase in participants in the norm-based intervention than in those in the contest-based intervention, we conducted a 2 (intervention: contest versus norm) × 2 (attitude: pre-versus post-measure) mixed ANOVA with repeated measures on the last condition on ratings of electricity conservation attitudes. A sensitivity power analysis, computed at $a = 0.05$, $1 - b = 0.80$, $r_{repeated\ measures} = 0.35$, $e = 1$, showed that the analysis was adequately powered to detect an effect size of $f = 0.39$. Results showed a tendency for an interaction ($F(1, 17) = 3.79$, $p = 0.068$, $\eta_p^2 = 0.18$). In line with H4, simple slopes effects found that attitudes increased in the norm-based intervention ($M_{dif} = 0.97$, $SE = 0.42$, $p = 0.03$) but not in the contest-based intervention ($M_{dif} = 0.09$, $SE = 0.36$, $p = 0.80$). This finding may explain why participants in the norm-based intervention showed a more long-lasting effect of electricity conservation, that is, by internalizing the interventions into their own attitudes. Behavioral change in participants in the norm-based intervention may have been motivated intrinsically (by attitudes toward electricity conservation) rather than extrinsically (by winning a prize). This finding, and the manipulation-check, indicates that participants in the norm-based intervention felt more cooperative while participants in the contest-based intervention felt more competitive, indirectly supporting our goal framing hypothesis.

Study 1 was limited by a small sample size and lack of a control group. Therefore, we conducted a second field experiment (Study 2) that aimed to recruit a larger sample by offering an increased number

of people the possibility to participate and also sending them a reminder to participate. Surveys were further undertaken at three time points, before, immediately after, and one month after the intervention. Study 2 also measured non-targeted pro-environmental behaviors. Research on spillover effects has found electricity conservation interventions to result either in negative spillovers such as increased water usage [51] and positive spillovers such as reduced beef consumption, sustainable transportation, and acceptance of pro-environmental policies [14,50], see also [52]. To examine whether contest-based and norm-based interventions differ in such non-targeted behaviors, Study 2 measured self-reported recycling and water conservation.

## 3. Study 2

### 3.1. Hypotheses

The aim of Study 2 was to replicate Study 1 while also improving the design. As in Study 1, we hypothesized that:

**H1.** *Participants in the contest-based intervention condition will show more intensive but shorter-lived electricity saving than participants in the norm-based intervention condition.*

**H2.** *Participants in the norm-based intervention condition will show less intensive but more long-lasting electricity saving behavior than participants in the contest-based intervention condition.*

**H3.** *Participants in the norm-based intervention condition will show more increase in personal electricity conservation norms than participants in the contest-based intervention condition.*

**H4.** *Participants in the norm-based intervention condition will show more increase in positive electricity conservation attitudes than participants in the contest-based intervention condition.*

Based on research suggesting that a normative goal framing positively affects non-targeted behavior [13,41,50] we added the following hypotheses:

**H5a.** *Participants in the norm-based intervention condition will increase their recycling more than participants in the contest-based intervention.*

**H5b.** *Participants in the norm-based intervention condition will increase their water conservation more than participants in the contest-based intervention.*

### 3.2. Method

**Procedure.** We offered 780 student apartments in Umeå, Sweden the opportunity to participate in an electricity conservation study. The overall distribution of type of apartments were: one-room apartments (17.4%) two-room apartments (62.4%) three-room apartments (17.1%) and four-room apartments (3.1%). As in Study 1, apartments were recruited via door hangers stating that the study would run for four weeks and that participation would be compensated with two lottery tickets. To increase participation rate, apartments also received a reminder in their mailbox (see Table A1 for overview of the experimental design). All subjects were informed that participation was anonymous, that they had the rights to end their participation, and that the data would be used for research purposes only. The study was conducted in accordance with the Declaration of Helsinki. We measured participants' electricity conservation attitudes, personal electricity conservation norms, self-reported water usage, and recycling prior to, directly after, and one month after the intervention. We also obtained apartments' electricity usage one month prior to, during, and one and a half months after the intervention (1st September to 18th December, 2016).

**Control condition.** Three apartment blocks ($n_{block1}$ = 72, $n_{block2}$ = 75, and $n_{block3}$ = 23) whose residents were not invited to participate in the study were used as a control and their anonymous electricity data were provided from *Umeå energy* from 1 September to 18 December, 2016. A one-way ANOVA testing for kWh usage in the three apartment blocks was significant ($F(2,167) = 394.39$, $p < 0.001$).

Bonferroni corrected post-hoc comparisons revealed that apartment block #3 used significantly more electricity than both apartment block #1 and apartment block #2 ($p's < 0.001$). Electricity usage did not differ significantly between apartment block #1 and #2 ($p = 1$). Apartment block #3 was therefore excluded, leaving the control group with 147 apartments for the main analysis.

**Intervention.** As in Study 1, participants were exposed to the social norm or contest manipulation via posters put up in visible spaces in all apartment blocks, on the refrigerator magnet and in an online survey. In addition to the procedure in Study 1, a feedback e-mail was sent half-way through the intervention. This feedback e-mail reminded participants in the contest intervention that the student who saved the most electricity by the end of the study would win a gift certificate of 500 SEK. The prize was later rewarded to the winner. For participants in the norm intervention, the feedback e-mail included an injunctive norm via a graph showing that the majority of participants believed that saving electricity was the right thing to do. Participants in both interventions were informed that other people engaged in electricity conservation (based on data from the first survey). This information was expected to be interpreted as a descriptive norm for participants in the norm intervention, showing that others act in congruence with the injunctive norm. For participants in the contest intervention, the information that others engaged in electricity conservation was expected to stress that other people also were interested in winning the price, framing a gain goal.

**Measures.** In the first survey, attitudes were measured using the same items as in Study 1 ($M = 5.88$, $SD = 1.17$). Based on past research [5], personal electricity conservation norms were assessed with two items measured on a 7-point Likert scale: "Do you feel a strong personal obligation to save electricity?" (1 = "very little" to 7 = "very much") and "I think I ought to save electricity" (1 = "do not agree" to 7 = "completely agree"; $M = 5.36$, $SD = 1.53$). The non-targeted behaviors recycling and water conservation were assessed by two items: "How often do you recycle?" ($M = 5.94$, $SD = 1.09$) and "How often do you conserve water?" ($M = 4.24$, $SD = 1.37$) both measured on a 7-point Likert scale (1 = "never" to 7 = "always").

The second survey was identical to the first, except that it also included control questions, such as whether participants had put up the refrigerator magnet during the intervention, whether they had seen the norm and/or contest posters, and how competitive and cooperative they made participant feel, and whether they had received and remembered the content of the feedback e-mail. Participants in the contest intervention were also asked whether they knew what the prize for saving the most electricity was. The third survey was shorted, and did only contain questions assessing attitudes, personal norms, water conservation, and recycling.

*3.3. Results and Discussion*

**Control questions and reliability analysis.** Valid kWh data were obtained for 60 participants randomly assigned to either a contest-based (n = 30) or a norm-based intervention (58.8% females, $M_{age} = 25.83$, $SD = 4.38$, recruitment rate = 7.69%). Of those 60 participants, 41 reported putting up the refrigerator magnet ($n_{contest} = 21$), 46 reported seeing the posters during the intervention ($n_{contest} = 27$), and 34 reported having read the feedback e-mail ($n_{contest} = 16$). As a control question, participants were probed for their degree of perceived cooperativeness and competitiveness when they saw the poster manipulation. Participants in the contest intervention reported feeling more competitive than participants in the norm intervention ($p = 0.018$, $d = 0.68$, 95% CI [0.12, 1.25]). No significant difference in cooperation was found between the interventions ($p > 0.05$). Cronbach's alpha analysis showed acceptable to high reliability values for the attitude items in the first ($\alpha = 0.77$), second ($\alpha = 0.86$), and third time of measure ($\alpha = 0.89$). The Cronbach's alpha for personal norms also showed acceptable reliability values for the first ($\alpha = 0.80$), second ($\alpha = 0.73$), and third time of measure ($\alpha = 0.71$).

**Main analysis.** We performed a 3 (intervention: contest versus norm versus control) × 3 (change in kWh usage compared to pre-intervention period: first two weeks versus last two weeks versus post-intervention) mixed ANOVA with repeated measures on the last condition tested for relative change in kWh usage. A sensitivity power analysis, computed at $a = 0.05$, $1 - b = 0.80$,

$r_{\text{repeated measures}} = 0.98$, $e = 1$, showed that the analysis was adequately powered to detect an effect size of $f = 0.02$. Results revealed a main effect of change in kWh usage ($F(2, 204) = 4.25$ $p = 0.02$, see Table 2 and Figure 2), showing that electricity usage increased across interventions. Testing the interaction between interventions and electricity usage, simple slopes analysis found that those in the contest intervention used significantly less electricity during the first two weeks of the intervention than those in the control condition ($p = 0.04$), and marginally less electricity than those in the norm intervention ($p = 0.06$). Participants in the norm-based intervention did not however save more electricity than those in the control condition during either the intervention or the post-intervention ($p > 0.05$). We were thus unable to replicate this finding from Study 1 and H2 was not supported. Testing the overall trend of electricity usage, within-subject contrasts found a significant quadratic trend ($F(1, 29) = 6.89$, $p = 0.01$) for participants in the contest-based intervention. As shown in Figure 2, these data support H1 and replicates the finding from Study 1, showing intensive but short-lived electricity savings in the contest-based intervention. Electricity usage decreased by 10% for the first two weeks of the intervention, but re-stabilized at 1% for the last two weeks of the intervention.

**Table 2.** Means and standard deviations for kWh usage in Study 2.

|  | Pre-Intervention | Week 1 and 2 | Week 3 and 4 | Post-Intervention |
|---|---|---|---|---|
| Contest-based intervention | 4.07 (1.49) | 3.71 (1.53) | 4.08 (1.64) | 4.09 (1.53) |
| Norm-based intervention | 4.02 (1.50) | 4.18 (1.79) | 4.24 (1.69) | 4.32 (1.80) |
| Control condition | 3.87 (1.64) | 3.86 (1.83) | 3.88 (1.55) | 4.04 (1.81) |

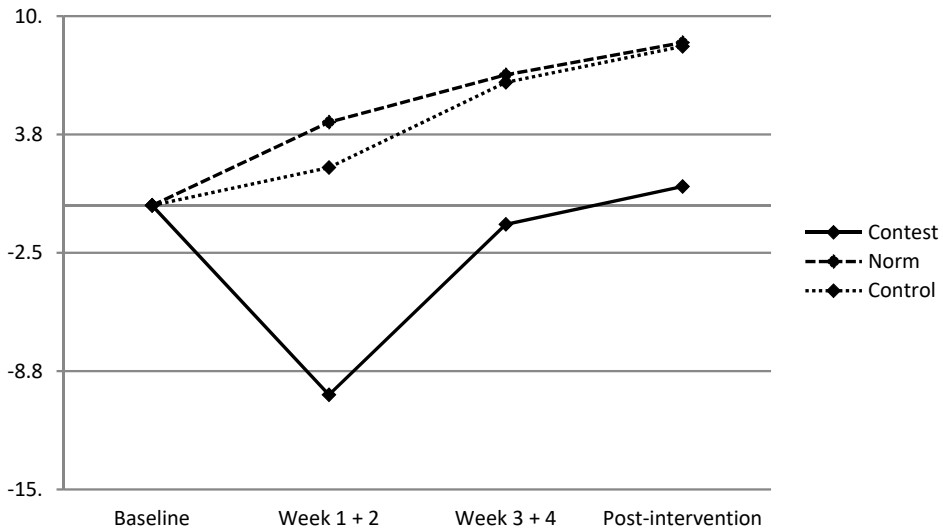

**Figure 2.** Percentage changes in kWh use from baseline to the first two weeks of the intervention, the second two weeks of the intervention and during the post-intervention period for the control condition and the contest-based and norm-based interventions.

Of the sixty participants recruited to Experiment 2 forty-eight individuals completed all three surveys ($n_{\text{contest}} = 19$), and were therefore included in the analyses to test hypotheses 3–5. To test H3, that participants' personal electricity conservation norms would increase more in the norm intervention than in the contest intervention, we conducted a 2 (intervention: contest versus norm) × 3 (time: pre-intervention versus intervention versus post-intervention) mixed ANOVA with repeated measures on the last condition on personal electricity conservation norms. A sensitivity power analysis, computed at $a = 0.05$, $1 - b = 0.80$, $r_{\text{repeated measures}} = 0.77$, $e = 1$, showed that the analysis was adequately

powered to detect an effect size of $f = 0.13$. Results showed no significant effect on personal electricity conservation norms ($p$'s $> 0.05$), therefore H3 was not supported.

To test H4, we conducted a 2 (intervention: contest versus norm) $\times$ 3 (attitudes: pre-intervention versus intervention versus post-intervention) mixed ANOVA with repeated measures on the last condition for ratings of attitudes toward electricity conservation. A sensitivity power analysis, computed at $a = 0.05$, $1 - b = 0.80$, $r_{\text{repeated measures}} = 0.55$, $e = 1$, showed that the analysis was adequately powered to detect an effect size of $f = 0.18$. Results showed a significant main effect of attitude ($F(1.56, 46) = 3.77$, $p = 0.04$, $\eta_p^2 = 0.08$). Simple slopes analysis showed that only for participants in the contest intervention were attitudes stronger during the intervention than prior to the intervention ($p = 0.001$), and after the intervention ($p = 0.01$), which was not predicted by H4.

Finally, H5a and H5b predicted a positive effect of electricity conservation on the non-targeted behaviors of recycling and water conservation in the norm intervention only. However, since participants in the norm intervention did not differ from the control condition in electricity conservation, our criteria for expecting such an effect was not fulfilled. Still, some participants in the norm intervention may have saved electricity, and for those the effect on non-targeted behaviors could still be expected. We therefore tested whether type of intervention moderated a positive relationship between both electricity conservation and recycling, and electricity conservation and water conservation. We computed three index variables measuring change in kWh usage and change in self-reported water usage and recycling using the measurement prior to and after the intervention. To test H5a and H5b, we ran two moderation analyses in *process* for SPSS [53]. In two ordinary least square multiple regressions, recycling and water conservation were regressed on electricity conservation; intervention was dummy coded (Contest = 0, Norm = 1) and used as a moderator variable. The first process model for H5a did not reach significance $F < 1$, rejecting H5a. When testing H5b, we found a significant model ($F(3, 53) = 3.24$, $p = 0.03$, $R^2 = 0.15$), and a significant interaction ($\beta = 1.0$, $SE_\beta = 0.49$, $t(57) = 2.1$, $p = 0.04$) supporting H5b by showing a significant positive relationship between electricity and water conservation in the norm intervention ($\beta = 0.58$, $SE_\beta = 0.27$, $t(29) = 2.15$, $p = 0.04$), but not in the contest intervention ($\beta = -0.43$, $SE_\beta = 0.41$, $t(29) = -1.1$, $p = 0.30$, see Figure 3).

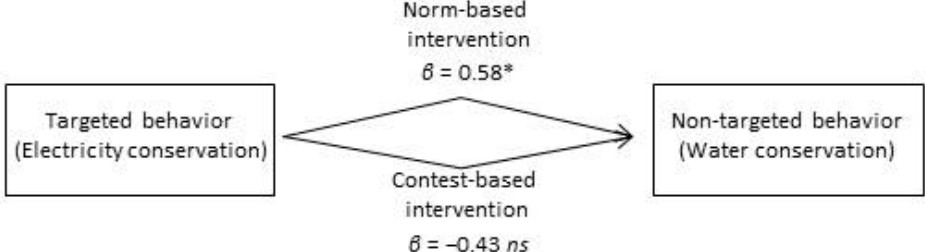

**Figure 3.** A positive relationship between electricity conservation and water conservation was supported in the norm-based intervention, suggesting that the non-targeted pro-environmental behavior water conservation was positively effected in the norm-based intervention but not in the contest-based intervention. Note *ns* = not significant ($p > 0.05$), * = $p < 0.05$.

## 4. General Discussion

We began the article by asking how intervention techniques may change behavior by inducing different goals. Drawing on Goal Framing Theory (GFT), [37], we argued that a contest-based intervention is a form of gain goal frame, while a norm-based intervention is a form of normative goal frame. The findings from the two field experiments partially support this proposition.

Participants assigned to a contest reduced their electricity usage by 20% in Study 1 and 10% in Study 2. However, this intensive behavioral change only persisted for the first half of the intervention period. Although these findings support our predictions, the data actually showed that electricity conservation motivated by the contest was even more short-lived than expected, as the effect did not last throughout the intervention period. We hypothesized that this short-lived effect was due

to a more instrumental view of electricity conservation, as participants would be motivated to save electricity only for as long as it was associated with a possible gain. From a psychological perspective, the contest-based intervention could thus be described as an extrinsic motivator [8,54–56], meaning that behavioral change is externally rather than internally motivated (see [8] for a review). In line with psychological consequences of externally motivated behavioral change, the contest resulted in short-term behavioral change [25]. In contrast, behavioral change in the norm-based intervention is likely to be attributed internally, as people tend to underestimate that they are in fact affected by external information in normative influences [57–59].

As a further consequence of framing a gain goal, participants in the contest showed no change in perceived electricity conservation norms in either Study 1 or in Study 2. Similarly, electricity conservation attitudes were not affected in Study 1, while an unexpected positive change in attitudes was found in Study 2. Following up on this unexpected finding, we found that although attitudes increased during the intervention compared with before the intervention ($p < 0.001$), attitudes regressed to baseline after the intervention ($p = 0.77$). These data are in line with our general proposition that contests have instrumental effect. Finally, in line with our hypotheses, no positive change in the non-targeted behaviors water conservation or recycling was obtained in the contest intervention. In general, these findings complement the electricity data in suggesting that the contest-based intervention is a form of gain goal frame.

For the norm-based intervention, we hypothesized a normative goal framing process in which electricity conservation would foster pro-environmental attitudes. In line with this prediction we found that engaging in a norm-based intervention increased positive electricity conservation attitudes in Study 1, but Study 2 did not replicate these findings. In Study 2 we found that only participants in the norm-based intervention showed a positive relationship between electricity and water conservation. This finding has both practical and theoretical implications, suggesting that only norm-based interventions may promote non-targeted behavior (i.e., positive spillover effects) [60]. In line with GFT [13,41], we suggest that this can be explained by the norm-based intervention framing a normative goal that activated a general goal to act "appropriately", which not only affects electricity saving but also other conservation behaviors (in this case water saving). It is important to note the difference between how we tested and analyzed attitudes and personal norms and how we tested non-targeted behaviors. In the latter case, we found a moderately strong relationship between electricity conservation and water conservation, showing a positive correlation only in the norm-based intervention. Increased engagement in one pro-environmental behavior area was thus related to increased engagement in another. However, attitudes and personal norms were tested as mean difference. One explanation for the lack of effect on personal norms is therefore that participants may need to engage more in the intervention for personal norms to increase. Moreover, it has been reported that norm-based interventions are less influential in people who already have high personal norms [28]. Personal norms might thus moderate behavioral change in norm-based interventions in people who do not already hold high personal electricity saving norms. We encourage future research to examine the conditions under which norm-based interventions affect behavior, and whether normative goal framing can provide a basis for positive spillover effects.

To explain why the predicted electricity usage was consistent for the contest-based intervention but not for the norm-based intervention it is important to compare the two studies. Our control questions showed that 95% of the participants in Study 1 reported putting up the refrigerator magnet and had seen the posters during the intervention while in Study 2, only 68% had put up the refrigerator magnet and 77% had seen the posters. This may be because approximately half of the posters were mistakenly taken down in Study 2. (A re-analysis of the data found no differences between groups where posters were taken down compared to groups where posters where not taken down) Moreover, only 57% of the participants in Study 2 reported that they had read the feedback e-mail. These data seem to suggest better data quality in Study 1 than in Study 2.

An important strength of Study 1 was that the apartments were highly homogeneous, while apartments were less homogeneous in Study 2. Hence, differences in electricity usage may have been confounded by type of apartment in Study 2, while apartments were less diverse in Study 1. Another difference in the norm-manipulation between the two studies is that Study 1 appealed to a more local norm than did Study 2. In Study 1, we informed participants about "other students at Lindholmsallén" (street of residence), but in Study 2 we used the wording "other students in Umeå" (town of residence). This discrepancy in manipulation was due to practical reasons, as we had no way of knowing which apartment blocks in Umeå would participate. However, there is evidence that the injunctive norm is especially sensitive to the use of local reference groups, while more general reference groups seem to be less influential [61]. In line with this explanation, participants in the norm intervention in Study 1 reported to feel more cooperative than participants in the contest intervention ($d = 0.29$). Interestingly, no such tendency was found in Study 2. This indicates that the norm-manipulation was less successful in Study 2, and it may explain the lack of effect on electricity conservation for the norm intervention.

As a general limitation, it should be noted that the recruitment rate was low in both Study 1 (12.67%) and Study 2 (7.69%). Although we randomly assigned these participants to one of the two interventions, our sample may differ from the general population. Our studies might for example have attracted students that were already interested in electricity conservation. If this is the case, participants may have engaged in electricity conservation before the interventions restricting the possibility for further electricity saving. Previous energy conservation interventions have reported recruitment rates of 19%, 18%, and 15%, and final samples of 19%, 6%, and 4% [19,27,62]. The recruitment rate in Experiment 1 was similar to these studies (14.6%), but lower in Experiment 2 (7.7%). When comparing our final samples to past energy conservation interventions, our samples were in line with what could be expected (12.67% and 7.67% / 5.26%). Still, this indicates that response rates might generally be low for interventions that target private and high-cost behavior. A related limitation is that both studies were based on small sample size. This restricts the validity of the present findings. It is however important to emphasize that we conducted two field-experiments, replicating for example the short-termed effect of contest-based interventions, which strengthens the validity of the results. Scaling up the studies, both in terms of number of participants and recruitment rates is very important. This is not what we are exploring in this study, but worth considering for future research. While recruitment levels may limit the generalizability of intervention studies, our student sample doesn't necessarily differ from the general population with regards to the antecedent motivation for these interventions. While conformity to social norms do increase up until the age of 18 years, level of conformity appears to be stable between the ages of 18 to 30 [63]. Therefore, we would not expect our samples' level of conformity to differ from a general (younger) population, but may differ from older populations. It's however possible that students' perceive their group as more homogenous, and therefore are more influenced by normative information.

The student sample may also differ in other ways that may influence the results. With regards to the contest-based intervention, students may have more limited financial resources than other segments of the population. However, we argue that as the contest provides external motivation for behavior change, the degree of change would depend on the level of incentive offered. Thus, the extent to which the results from the contest intervention generalize to the general population would depend on the level of incentive offered. The sample of households in our studies differs from a general population as student households consist of more single households, include no (or very few) children, and are normally occupied during less of the time. We encourage future research to investigate other populations to discern possible differences to our student sample.

Electricity conservation in households may be an important part in reducing $CO_2$ emissions. Over the past two decades, the electricity efficiency of household appliances has improved substantially, while domestic electricity demand has increased in the same period. This apparent paradox can be attributed to an increase in electricity-demanding equipment. The potential to reduce domestic electricity use is however relatively good. For example, in one study [1] the behavioral plasticity

(maximum potential for energy reduction through behavioral change) for the use of standby equipment and laundry behaviors was estimated at 35%. Many of these changes in behavior are also relatively easy for individual households to achieve and may be substantial in terms of reduced $CO_2$ emissions. For example, in Study 1, the average electricity conservation was 12.9% below baseline during the intervention, and 9% during the post-intervention period for the two interventions.

Social comparisons are now being applied in large scale interventions [29], and practitioners often use contest-based interventions (e.g., "Student Switch Off") to motivate behavioral change. These interventions, however, are seldom formed or evaluated in terms of motivations and goals. As a general recommendation, the present studies highlight the importance of investigating relevant motivators and goals to change behavior. Interventions designed to increase electricity conservation will activate both of these motivations and goals, and without appropriate knowledge about these interactions, interventions may fail or even backfire. The use of theory is essential in this endeavor, as it will enhance the prospect of learning from previous interventions, and allow comparison of results from one context to another. Our studies, in line with previous research investigating the effects of external rewards on electricity conservation [19], found the effects of external rewards to dissipate after two weeks. External motivation may thus be less effective and even more short-lived than previously thought, at least when the external rewards are given in the form of prizes. These results can be useful for policy makers, utility operators, and others who need to assess demand management through different instruments.

On a more specific level, contest-based interventions may best be used when aiming to promote intense but short-term engagement, while norm-based intervention techniques may best be used when aiming to promote possible long-term effects. For curtailment behaviors, which are typically repeated over a long period, using social norms may be a better option. Contests, on the other hand, may be more suitable for one-shot behaviors, since contests may be more likely to promote an intense but short-lived motivation to conserve electricity.

## 5. Conclusions

The aim of the present studies was to compare a contest-based intervention to a norm-based intervention. Results from two field-studies suggest that a contest-based intervention motivates behavioral change through the goal to gain or save money, while the norm-based intervention motivates behavior change though the goal to act appropriately. The studies found that both contest-based and norm-based interventions have the ability to promote electricity conservation. The behavioral change induced in the contest was more intensive and less long-lived than the change in the norm-based intervention. Only in the norm-based intervention was increased electricity conservation positively related to increased water conservation (i.e., a positive spillover effect). These findings emphasize the importance of understanding the motivational antecedents in different intervention techniques, having both theoretical and practical implications for promoting pro-environmental behavioral change.

**Author Contributions:** M.B. and A.N. developed the study concept and study design. All authors contributed in designing survey and stimulus material. The data was collected and analyzed by M.B. and E.E. All authors contributed in writing, editing, and reviewing the manuscript.

**Funding:** Please add: "This research was funded by the Swedish Energy Agency, grant number 40242-1, and Centre for collective action research (CeCAR)".

**Acknowledgments:** The authors would like to thank Trevor Archer and Juliane Buecker for valuable comments on the manuscript. The authors would also like to thank Studentbostadsföretagen, SGS Studentbostäder, Göteborgs energi, AB Bostaden, Umeå energi, and Meike Ortmanns for helping in collecting and providing the data.

**Conflicts of Interest:** The authors declare no conflict of interest.

## Appendix A

**Table A1.** Summary of experimental design in Study 1 and Study 2.

|  | Four Weeks before the Intervention | Two Weeks before the Intervention | Recruitment | First Day of the Intervention | Week 1 and 2 | Week 3 | Week 4 | Week 5 | Week 6–8 |
|---|---|---|---|---|---|---|---|---|---|
| Study 1 | - | kWh-usage was measured two weeks before the intervention. | 150 apartments were invited using door hangers. | Intervention started. Prompts were put up, survey 1 was emailed, and refrigerator magnets were given to the participants. | Intervention period (n = 19). | Intervention ended. Survey 2 was emailed. | Ending kWh-measuring after week 4. | - | - |
| Study 2 | kWh-usage was measured one month before the intervention. | Measuring kWh-usage. | 780 apartments were invited using door hangers and a reminder in their mailbox. | Intervention started. Prompts were put up, survey 1 was emailed, and refrigerator magnets were mailed to the participants. | Intervention period (n = 60 + 147). | Intervention period. Survey 2 and feedback was e-mailed. | Intervention period. | Intervention ended. Survey 3 was emailed at week 5 (n = 41). | Ending kWh-measurement at week 8. |

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
