# Peer review of "Contest-Based and Norm-Based Interventions: (How) Do They Differ in Attitudes, Norms, and Behaviors?"

_sustainability, doi:10.3390/su11020425_

Round 1
Reviewer 1 Report
Interesting paper, in general well written and dense of information.
My main concern is the sample size, the numerous statistical analyses and the thorough discussion of the results are performed on the basis of a sample that can hardly support a 2-group t-test.
The feeling, after reading the whole paper, is that perhaps it is much ado about nothing.
The authors should at least include a power analysis to convince the reader that the group comparisons make sense.
Author Response
We would like to thank all reviewers for the comments. We have made an effort to amend the issues raised in the reviews. Please see attachment:

Reviewer 2 Report
The aim of the paper is to report two studies of behavioral and psychological effects of contest-based and norm-based intervention techniques for promoting electricity conservation. Results from both studies showed that the contest-based intervention promoted intensive but short-lived electricity savings. Participants in the norm-based intervention also engaged in other types of sustainable behavior than the targeted – in addition to electricity conservation participants also engaged in water saving activities. This suggests that norm-based, but not contest-based interventions, promote positive spillover effects.
The article draws upon the theory of Goal Framing Theory (GFT), which is a relevant theory. However, it would be desirable for the readers to learn more about the practical application of the theory. What kind of everyday phenomena does it set out to explain?
As the concepts of norms and attitudes are central to the article, they need to be described in greater detail also specifying the difference between them. The authors need to consider that the audience of Sustainability is multi-disciplinary. The difference between the concepts “descriptive norm” and “injunctive norm” also needs to be described more extensively. Also, it would be enlightening to be given more examples of normative goals and hedonic goals (line 74-75).
The study suffers from small sample sizes, which is also pointed out by the authors. This, however, weakens the conclusions drawn from the study although a conclusion of trends may be drawn.
The statistical reporting is often too detailed (e.g. “Main Analysis”, starting on line 208) for the multi-disciplinary audience of the journal. It may be better to put the statistics in a table and explain it in the text. The readers may thus choose to read the details in the table if they’d like to. The statistical concepts “simple slope analysis” (line 213, 223) and “quadratic trend” (line 220) may not be familiar to the readers. Neither may the readers be familiar with the terminology of experimental psychology, such as that described on line 243-245.
Overall, the article’s style has a within-disciplinary focus (psychology). For instance, the audience is expected to understand the statistical terminology which might not be the case for the multi-disciplinary audience of the journal.
The authors state that practitioners often use contest-based interventions (line 38, line 550), but no examples or illustrations of these are given.
Author Response

(The authors gave the same response as above.)

Reviewer 3 Report
The paper addresses relevant themes on way for engaging people in changing possibly energy-consuming behaviours by analysing and veryfing two main strategies (norm and contest based ones).
The overall hypotheses are eventually tested with student population in specific halls. Although the authors acknowledge such a limitation in a concluding section, this element results of fundamental importance to freme the hypothesis and methodology. Interest in savings and rewards, together with will to follow norms are enacted plausibly in a significantly different way between student population and other segments (e.g. workers, elders, unemployed).
The investigation and the paper as a consequence would appear more scientifically sound and consistent if the hypothesis is built upon the engaged sample. My suggestion is therefore to reframe the paper based on this and the demonstrating the importance of intervening in/with/for this population segment due to their energy consuming practices and specific ways of responding to norm-based and contest-based initiatives.
Also, the limited number of participants in the fieldwork activities is typical of qualitative rather than quantitative investigations. A different approach to data analysis could provide more relevant information which may inform future quantitative oriented investagations.
Other comments are included in the lines of the attached file.

Author Response

(The authors gave the same response as above.)

Round 2
Reviewer 1 Report
The authors have clarified the many limitations of the study. I am satisfied with their explanation.
Author Response
Thank you very much for your comments and for taking your time to review our manuscript.
Reviewer 2 Report
The authors have improved the manuscript mostly in agreement with reviewers’ comments, although the sentence added regarding comment 3 (p. 3, line 125-128) still does not clearly distinguish between people’s attitudes and normative considerations. Viewing a specific conservation behavior in a positive or negative way (which the authors describe as people’s own opinions) doesn’t seem very different from perceiving “that it is right to conserve electricity”. If there is a central difference, what is it? Is it the generality and internalization of a guiding principle vs an attitude towards some specific behavior in a specific context? The authors refer to references 36 and 39 to clarify the difference, but this is not enough.
The study still suffers from small sizes, which weakens the conclusions. However, this is pointed out by the authors.
There are some minor language mistakes, e.g. the sentence L619-L621Author Response
We have now made changes throughout the manuscript to correct the misstakes pointed out by you and to enhance readability.
We have also added [122-124] the following sentence to clearify the difference between attitudes and normative considerations
"Although these concepts overlap, the key difference is that while attitudes measure people’s evaluations, normative considerations is intended to capture people’s moral beliefs."
Thank you very much for your comments and for taking your time to review our manuscript.